# Composite Membrane Based on Melamine Sponge and Boehmite Manufactured by Simple and Economical Dip-Coating Method for Fluoride Ion Removal

**DOI:** 10.3390/polym15132916

**Published:** 2023-06-30

**Authors:** Han-Bi Lee, Ah-Jeong Choi, Young-Kwan Kim, Min-Wook Lee

**Affiliations:** 1Institute of Advanced Composite Materials, Korea Institute of Science and Technology, Jeonbuk 55324, Republic of Korea; 092414@kist.re.kr; 2Department of Chemistry, Seoul Campus, Dongguk University, 30 Pildong-ro, Seoul 04620, Republic of Korea; caj9038@gmail.com

**Keywords:** melamine sponge, boehmite, adsorption, composite membrane

## Abstract

The wastewater generated from the semiconductor production process contains a wide range and a large number of harmful substances at high concentrations. Excessive exposure to fluoride can lead to life-threatening effects such as skin necrosis and respiratory damage. Accordingly, a guideline value of fluoride ions in drinking water was 1.5 mg L^−1^ recommended by the World Health Organization (WHO). Polyvinylidene fluoride (PVDF) has the characteristics of excellent chemical and thermal stability. Boehmite (AlOOH) is a mineral and has been widely used as an adsorbent due to its high surface area and strong adsorption capacity for fluoride ions. It can be densely coated on negatively charged surfaces through electrostatic interaction due to its positively charged surface. In this study, a composite membrane was fabricated by a simple and economical dip coating of a commercial melamine sponge (MS) with PVDF and boehmite to remove fluoride ions from semiconductor wastewater. The prepared MS-PVDF-Boehmite composite membrane showed a high removal efficiency for fluoride ions in both incubation and filtration. By the incubation process, the removal efficiency of fluoride ions was 55% within 10 min and reached 80% after 24 h. In the case of filtration, the removal efficiency was 95.5% by 4 cycles of filtering with a flow rate of 70 mL h^−1^. In addition, the removal mechanism of fluoride ions on MS-PVDF-Boehmite was also explored by using Langmuir and Freundlich isotherms and kinetic analysis. **(R2-1)** From the physical, chemical, thermal, morphological, and mechanical analyses of present materials, this study provides an MS-PVDF-Boehmite composite filter material that is suitable for fluoride removal applications due to its simple fabrication process, cost-effectiveness, and high performance.

## 1. Introduction

Wastewater from the semiconductor manufacturing process contains a wide range of harmful substances at high concentrations. Among them, fluoride ions are particularly important because they are released at high concentrations not only from semiconductor manufacturing but also from coal-fired power plants, which contaminate groundwater. The concentration of fluoride ions discharged from these industrial activities can reach several hundred to several thousand mg L^−1^. When it presents in drinking water at low concentrations, fluoride ions can prevent tooth decay, but if they present at high concentrations higher than 1.5 mg L^−1^, they can cause fluorosis in bones and teeth, making them a hazardous substance. In addition, if more than 5 mg of fluoride ions per 1 kg of body weight are consumed excessively, gastrointestinal disorders, nausea, vomiting, and in severe cases, death can occur [1]. Accordingly, the Centers for Disease Control and Prevention (CDC) in the United States recommends fluoride ion concentrations in drinking water of 0.7–12 mg L^−1^, while the World Health Organization (WHO) recommends a guideline value of 1.5 mg L^−1^ [1,2,3,4].

There are various methods available for removing fluoride ions from wastewater. Representative methods include chemical precipitation, ion exchange resin, membrane filtration, adsorption, and precipitation. The chemical precipitation method is a method of precipitating CaF_2_, an insoluble salt, by neutralizing calcium ions and fluoride ions by adding lime water [5,6]. However, this method is only effective for treating high concentrations of fluoride ions and cannot remove them to levels below 10 mg L^−1^. The ion exchange resin method removes dissolved fluoride ions by an ion exchange process [7], but it has the disadvantage of being unable to remove highly concentrated fluoride ions and having a high treatment cost. Membrane filtration has a high removal efficiency for fluoride ions [8,9], but it also has disadvantages such as high maintenance cost, fouling around the membrane, and a complex treatment process. In contrast, the removal of fluoride ions by adsorption has the advantages of requiring less energy and cost than other removal technologies and having a simple treatment process [10,11,12]. The adsorbents commonly used for fluoride ions include aluminum-based adsorbents, calcium-based adsorbents, hydroxides, boehmite, graphite, activated carbon, and others [13,14,15,16,17,18]. **(R2-2)** If it is difficult to use this adsorbent independently, and it may contribute to adsorption by immobilizing it on a membrane [19,20]. Materials for the membrane include polyacrylonitrile (PAN), polyvinylidene fluoride (PVDF), and polytetrafluoroethylene (PTFE) [21,22,23]. As for related studies, a separator that adsorbs copper ions by grafting PAMAM on the surface of a PVDF membrane has been reported [24], and a MOF membrane for removing Cd and Zn prepared by electrospinning of Zr-based MOF-808 and hydrophilic PAN has been reported [25]. These materials are suitable for aqueous applications and exhibit physical and chemical stability. However, we studied boehmite and sponge-based composite membranes, including higher adsorption capacity and superior mechanical properties, in addition to the advantages of previously reported composite membranes.

Boehmite is synthesized by a hydrothermal method under high temperature and pressure by putting a solid reactant and solvent into an autoclave [26]. Boehmite has the advantages of being environmentally friendly and cost-effective, and it is widely used as an adsorbent due to its high surface area of about 448 m^2^ g^−1^ [27]. Additionally, it carries a positive charge, making it suitable for electrostatic adsorption of negatively charged compounds [28,29]. Melamine sponge (MS) is a commercially available material and is widely harnessed as a support for various adsorbents due to its inherent characteristics, including high porosity, high absorbency, wide surface area, and low density [30,31,32]. Additionally, depending on the coating materials, it can selectively exhibit hydrophilic and hydrophobic properties, which is beneficial to applications in water/oil separation and absorption [33,34,35,36]. In this study, a composite membrane of MS and boehmite was fabricated through a simple and sequential dip coating of MS in a solution of polyvinylidene fluoride (PVDF) and an aqueous suspension of boehmites. The resulting composite membrane of MS-PVDF-Boehmite was thoroughly characterized with analytical tools to reveal its structure and directly applied to the removal of fluoride ions under various conditions. The MS-PVDF-Boehmite composite membrane (8 cm^3^) showed a high removal rate of 60% from a solution containing 1–80 mg L^−1^ fluoride ions, and its adsorption characteristics were also investigated with Langmuir and Freundlich isotherms and kinetic analysis. Based on its high performance, the MS-PVDF-Boehmite composite membrane was inserted into a syringe and applied as a cartridge to remove fluoride ions from flowing wastewater. After 4 cycles of the purification process, the concentration of fluoride ions fell to below the WHO standard (1.5 mg L^−1^), and the removal efficiency was 95.5%. **(R2-2)** In this work, a composite membrane for removal of fluoride ions was developed by a simple dip-coating process. The results indicated that the MS-PVDF-Boehmite composite membrane is also an effective tool to purify wastewater contaminated with fluoride ions.

## 2. Experimental Section

### 2.1. Materials

**(R1-1)** Aluminum isopropoxide (AIP, ≥98%), N,N-Dimethylformamide (DMF, 99.8%), and acetic acid (≥99.7%) were purchased from Sigma-Aldrich (St. Louis, MO, USA). Ethyl alcohol (EtOH, 94.5%) was purchased from Samchun Pure Chemicals (Pyeongtaek, Republic of Korea). Polyvinylidene fluoride (PVDF, Kynar-761) was purchased from Arkema (Singapore). Hydrochloric acid (HCl, 37%) and sodium hydroxide (NaOH, 97%) were purchased from Daejung Reagent Chemicals (Siheung, Republic of Korea). A melamine sponge (MS) was purchased from BASF (Ludwigshafen, Germany).

### 2.2. Synthesis of Boehmite (γ–AlOOH)

Boehmite (γ–AlOOH) was prepared by a conventional sol–gel reaction of AIP. For the synthesis, 68 g of AIP was added to 300 mL of de-ionized (DI) water at 75 °C, and the aqueous solution of AIP was heated at 95 °C with stirring until the total volume of the solution became 200 mL through evaporation. Next, 3.1 g of acetic acid was added to the AIP solution in a drop-by-drop manner and stirred for 10 min. Finally, a hydrothermal reaction was carried out by using an autoclave at 150 °C for 6 h. During this hydrothermal reaction, AIP was transformed into boehmite crystals.

### 2.3. Manufacturing Process

Boehmite-based composite membranes were fabricated with two different processes (Figure 1). First, the melamine sponge (MS) was washed with flowing DI water and EtOH. After washing, the MS was immersed in the boehmite solution for 1 h. The resulting MS-Boehmite was washed 3 times with DI water and dried in an oven at 50 °C. MS-boehmite was prepared through a one-step dip-coating process (Figure 1a). PVDF was put into a DMF solution and stirred at a constant speed for 4 h at 50 °C to prepare a 5 wt% PVDF solution. The MS was immersed in the PVDF solution at room temperature for 2 h. Next, the sample was taken out, and the excess solution was shaken off and dried in an oven overnight. The resulting MS-PVDF was washed 3 times with DI water and dried in an oven at 50 °C. After drying, MS-PVDF was immersed in boehmite solution for 1 h. The resulting MS-PVDF-Boehmite was washed 3 times with DI water and dried in an oven at 50 °C. MS-PVDF-Boehmite was prepared by electrostatic interaction by sequential coating with negatively charged PVDF and positively charged boehmite to improve the adhesion and bonding strength between MS and boehmite (Figure 1b). **(R2-3)** Boehmite was synthesized through a hydrolysis reaction between DI water and AIP precursor followed by hydrothermal treatment (Figure 1c). The synthesized boehmite showed a typical white color, and it was well dispersed in DI water, forming a translucent suspension (Figure 1d,e).

### 2.4. Characterization

To evaluate the adsorption performance of a composite membrane for fluoride removal, fluoride wastewater (initial concentration: 5000 mg L^−1^) was diluted to prepare solutions of fluoride ions at 1, 5, 10, 20, 40, and 80 mg L^−1^. The adsorption test was conducted in two ways. First, MS-PVDF-Boehmite samples were cut into 2 × 2 × 2 cm^3^, placed in 20 mL fluoride ion solutions at various concentrations and stirred at 50 °C for 24 h. After that, 1 mL of the fluoride ion solutions was collected to measure the changes of fluoride ion concentrations. Second, MS-PVDF-Boehmite samples were cut into π × 0.6^2^ × 4 cm^3^ and inserted into a 5 mL syringe. Then, 5 mL of 20 mg L^−1^ fluoride ion solution was injected at flow rates of 30, 70, and 110 mL h^−1^ to examine the adsorption efficiency under different flow rates. This experiment was repeated in cycles until the concentration of fluoride ions became lower than that of the WHO standard (1.5 mg L^−1^).

The physical, chemical, thermal, morphological, and mechanical characteristics of the prepared MS, MS-Boehmite, and MS-PVDF-Boehmite were systematically investigated. An X-ray diffraction (XRD) pattern (Rigaku, Tokyo, Japan) of boehmite was obtained with a Rigaku X-ray diffractometer equipped with a Cu Kα source. Their thermal properties were explored using thermogravimetric analysis (TGA Q50, TA Instruments, USA). In the thermal tests, samples were placed in a ceramic pan at a constant heating rate of 10 °C min^−1^ within 40–800 °C under a nitrogen–air atmosphere at a flow rate of 90 mL min^−1^. The surface and interface of the specimen were observed using an optical microscope (VHX-900F, Keyence Corporation, Osaka, Japan) and a scanning electron microscope (SEM, Nova NanoSEM 450, FEI) at 15 kV. The functional group analysis of the samples was performed using FT-IR (Sinco, Seoul, Korea). The compression tests were conducted at a crosshead speed of 10 mm min^−1^ with a dimension of 2 × 2 × 2 cm^3^. The concentration of fluoride ions was measured using a fluoride colorimeter (HI-739, HANNA Instruments, Woonsocket, RI, USA).

The adsorption capacity was calculated by Equation (1) from the measured concentration of fluoride ions remaining in the solution:(1)$$q_{e}=\frac{\left(C_{0}-C_{e}\right)V}{W}\;$$
*q_e_*: Equilibrium adsorption amount adsorbed per unit g of adsorbent (mg g^−1^)*C*_0_: Initial concentration of fluoride ion (mg L^−1^)*C_e_*: Equilibrium concentration of fluoride ion in solution after adsorption (mg L^−1^)*V*: Volume of solution (L)*W*: Adsorbent Dosage (g)


The heavy metal removal efficiency Re (%) was obtained by Equation (2).
(2)$$Re\left(\%\right)=\frac{\left(C_{0}-C_{e}\right)}{C_{0}}\times 100\;$$

## 3. Results and Discussions

### 3.1. Characterization of Synthesized Boehmite (γ-AlOOH)

SEM images showed the morphological characteristics of rod-like boehmites with a length of few hundred nanometers and a diameter from 20 to 50 nm (Figure 2a). To confirm the successful synthesis of boehmite, an XRD pattern of the synthesized sample was obtained (Figure 2b). The XRD pattern exhibited characteristic diffraction peaks at 2θ = 13.75°, 28.25°, 38.35°, and 49.20°, which correspond to the (020), (120), (031), and (200) planes of boehmite, and those peaks verified that boehmite was successfully synthesized under our synthetic condition. Then, the zeta potential value of boehmite at various pH conditions was measured to examine its surface charges, and it showed a positive zeta potential ranging from 20 to 40 mV and a pH range from 3 to 8 (Figure 2c). This result implied that boehmite can maintain its positive charge for the electrostatic adsorption of negatively charged contaminants such as fluoride ions in diverse environments. **(R1-2)** The negative charge of boehmite was caused by the increase in the number of OH^-^ groups with the increase in pH and the decrease in zeta potential value [37].

### 3.2. Morphology of MS, MS-Boehmite, and MS-PVDF-Boehmite

Figure 3 showed SEM and EDX images of MS, MS-Boehmite, and MS-PVDF-Boehmite composite membranes. MS has a smooth surface morphology and an interconnected 3D network framework (Figure 3a). After coating with boehmite, MS-Boehmite exhibited boehmites coated on its surface (Figure 3b), and Al was detected by EDX mapping (3.40%) (Figure 3b inset). Compared to MS-Boehmite, the surface coverage of boehmites on MS-PVDF-Boehmite was considerably enhanced, and as a result, its surface was rougher than MS and MS-Boehmite and was composed of large boehmite crystals (Figure 3c). In addition, Al content of MS-PVDF-Boehmite (4.59%) was also higher than that of MS-Boehmite (3.40%) (Figure 3c inset). This indicates that PVDF played an important role as an adhesive layer for the electrostatic adsorption of positively charged boehmites on the surface of MS due to its negative charges.

### 3.3. Characterization of MS, MS-Boehmite, and MS-PVDF-Boehmite

Figure 4a shows the FT-IR spectra of MS, MS-Boehmite, and MS-PVDF-Boehmite. Compared with MS [38], new peaks appeared at around 3068 and 1060 cm^−1^ from MS-Boehmite and MS-PVDF-Boehmite (Figure 4a), and those peaks correspond to the O-H vibration of AlOOH and to the Al-O-Al symmetric bending vibration, respectively. A strong peak of C-H stretching vibration was observed at 1398 cm^−1^ from MS-PVDF-Boehmite, and there were also peaks located at 1280 and 1011 cm^−1^ corresponding to C-F bond vibrations (Figure 4a). In addition, the peaks at 1494 and 1328 cm^−1^ from C=N and C-N bonds of MS were weakened with sequential coating with PVDF and boehmite (Figure 4a). The FT-IR analysis confirmed that PVDF and boehmite were successfully coated on the surface of MS.

TGA analysis was performed to investigate the thermal stability of the prepared composite membranes. Figure 4b,c show the TGA and DTA curves of MS, MS-Boehmite, and MS-PVDF-Boehmite in a nitrogen atmosphere. The TGA curve of MS showed a rapid weight loss in the temperature range of 330 to 400 °C, which occurs when the HN-CH_2_-NH bond is broken [39]. The weight loss at higher temperatures is due to thermal decomposition of the triazine ring. MS-Boehmite maintains thermal stability up to 335 °C, which is 10 °C higher than that of MS. It implied that the thermal stability of MS was enhanced by coating with boehmite. Interestingly, MS-PVDF-Boehmite retained its thermal stability up to 353 °C, which is approximately 20 °C higher than that of MS-Boehmite. It can be inferred that due to the important role of negatively charged PVDF as a binder, a greater amount of positively charged boehmite was coated on MS-PVDF than MS, and thus its thermal stability increased.

### 3.4. Mechanical Properties of MS, MS-Boehmite, and MS-PVDF-Boehmite

The mechanical properties of MS, MS-Boehmite, and MS-PVDF-Boehmite were explored with compression tests to reveal the coating effect of boehmite with different methods (Figure 5). A compression test of 10 cycles was conducted with a 70% strain and a strain rate of 10 mm min^−1^ (Figure 5a–c). MS and MS-Boehmite showed a similar compressive stress of 22.7 and 32.8 kPa, respectively, while MS-PVDF-Boehmite exhibited a relatively high compressive stress of 65.0 kPa, which is nearly threefold higher than MS and twofold higher than MS-Boehmite. The highly enhanced compressive stress implied that boehmite reinforced the mechanical properties of MS, and this effect is augmented with the PVDF adhesive layer leading to a high surface coverage of boehmite. The compression stress of MS-PVDF-Boehmite was partially diminished after 10 repeated compression tests, but there was no significant damage, confirming their high durability for the practical application. It is also worthy to note that the decreased compression stress of MS-PVDF-Boehmite was still much higher than that of MS and MS-Boehmite.

### 3.5. Isothermal Adsorption Test

The adsorption performance of MS, MS-Boehmite, and MS-PDVF-Boehmite (2 × 2 × 2 cm^3^ in their dimension) was explored by incubating them in standard solutions having initial concentrations of 1, 5, 10, 20, 40, and 80 mg L^−1^ with constant stirring at 230 rpm. After 24 h of incubation, the adsorption membranes (MS, MS-Boehmite, and MS-PVDF-Boehmite) were retrieved and 1 mL of the residual solutions was collected to measure the concentration of fluoride ions. As shown in Figure 6a, MS exhibited a removal efficiency of over 67% at 1 mg L^−1^ and showed a removal efficiency of over 37–40% at other concentrations, indicating fluoride ions can be adsorbed on MS without coating of boehmite. In the case of MS-Boehmite, it showed 100% removal efficiency at 1 mg L^−1^. This high removal efficiency was derived from the high surface area and strong affinity of boehmites for fluoride ions. However, it was confirmed that the removal efficiency decreased with an increasing concentration of fluoride ions. In the case of MS-PVDF-Boehmite, fluoride ions at concentrations of 1–10 mg L^−1^ were completely removed within 24 h, and the removal efficiency was about 78% or higher even at 20 mg L^−1^. The enhanced removal efficiency clearly indicated that the removal of fluoride ions was derived from the electrostatic adsorption of fluoride ions on the surface of boehmites, and thus the removal efficiency significantly increased with the loading amount of boehmites. In addition, when the pH value was less than 5.0, hydroxyl groups on the surface of boehmites were prone to be protonated for the formation of −$OH_{2}^{+}$ in acidic solutions. Therefore, the surface of boehmites became further positively charged and facilitated the electrostatic adsorption of fluoride ions [29,40].

The adsorption capacity (mg g^−1^) of MS, MS-Boehmite, and MS-PVDF-Boehmite for fluoride ions was examined with different initial concentration (Figure 6b). The adsorption capacity was determined to be 2.98, 2.60, and 1.06 mg g^−1^ for MS, MS-Boehmite, and MS-PVDF-Boehmite at 20 mg L^−1^ of fluoride ions, respectively. Interestingly, although MS-PVDF-Boehmite has the highest removal efficiency of fluoride ions among the tested samples, its adsorption capacity was significantly lower than that of MS and MS-Boehmite. This low adsorption capacity was attributed to its increased weight compared to MS and MS-Boehmite because MS, MS-Boehmite, and MS-PVDF-Boehmite were prepared in an equal dimension (2 × 2 × 2 cm^3^), and thus MS-PVDF-Boehmite has the highest weight among the tested samples due to the PVDF adhesive layer and high loading amount of boehmites (the weight of MS-PVDF-Boehmite was fivefold higher than that of MS). To quantitatively compare the adsorption capacity, MS, MS-Boehmite, and MS-PVDF-Boehmite were cut with an equal weight (0.07 g) and different dimensions such as 2.0 × 2.0 × 2.0, 1.7 × 1.7 × 1.7, and 1.2 × 1.2 × 1.2 cm^3^, respectively. The removal efficiency and adsorption capacity of MS-PVDF-Boehmite were 27.1% and 1.5 mg g^−1^ and these values were still lower than those of MS (43.8% and 2.4 mg g^−1^) and MS-Boehmite (34.8% and 1.9 mg g^−1^) **(R2-4)** (Figure 6e). Considering a nearly fivefold smaller volume of MS-PVDF-Boehmite than MS, its adsorption performance is sufficient for the practical application of removing fluoride ions.

Then, the adsorption mechanism of fluoride onto MS, MS-Boehmite, and MS-PVDF-Boehmite was investigated with Langmuir and Freundlich isotherm models. These models are extensively harnessed to study a solid–liquid interface system at adsorption equilibrium. To determine the suitability of each isotherm model, three error functions such as coefficient of determination (R^2^), sum of absolute error (SAE), and chi-square (χ^2^) were calculated from each isotherm model, respectively (Table 1).

The Langmuir isotherm equation indicates that the adsorption is mainly conducted by the bonding force between the surface of adsorbents and aqueous adsorbates. Therefore, the Langmuir model assumes that adsorbate forms a monomolecular layer onto the adsorbents without lateral interactions, and no further adsorption occurs when monolayer adsorption is completed [41]. The nonlinear form of the Langmuir isotherm model can be expressed as Equation (3):(3)$$q_{e}=\frac{q_{max}K_{L}C_{e}}{1+K_{L}C_{e}}$$

Here, the *K_L_* is Langmuir constant, which is a crucial parameter that can determine the adsorption rate (L mg^−1^), and *q_max_* is the maximum adsorption capacity (mg g^−1^) for fluoride ions, representing the theoretical maximum monomolecular layer adsorption capacity of the used adsorbents.

The Freundlich adsorption isotherm is a semi-experimental model derived from the Langmuir isotherm. It implies multilayered adsorption with uneven distribution of adsorption energy on the surface of adsorbents. It assumes that adsorbates are initially adsorbed on the stronger adsorption site of adsorbents, and the adsorption heat decreases gradually with increasing coverage of active sites of adsorbents. The nonlinear Freundlich isotherm equation is expressed as Equation (4):(4)$$q_{e}=K_{F}C_{e}^{\frac{1}{n}}$$

Here, *K_F_* is the Freundlich constant related to adsorption capacity of the adsorbent (L mg^−1^), and n is a measure of adsorption intensity, which can vary with the surface heterogeneity and affinity of adsorbents. A higher *K_F_* value indicates a better relative adsorption capacity [42]. The experimental adsorption data were fitted using Langmuir and Freundlich models, as shown in Figure 6c,d. The calculated adsorption isotherm parameters and error functions from the two models are summarized in Table 2.

The *q_max_* values of MS, MS-Boehmite, and MS-PVDF-Boehmite were obtained as 6.36, 9.47, and 1.58 mg L^−1^ by using the Langmuir isotherm, respectively. This result is consistent with the experimental adsorption results, which showed that MS-PVDF-Boehmite possessed the lowest adsorption capacity due to the increased density of the sponge samples. Using the Freundlich isotherm, the *K_F_* values of MS, MS-Boehmite, and MS-PVDF-Boehmite were calculated to be 0.528, 0.704, and 0.837 L mg^−1^, respectively. Those results suggest that the adsorption capacity of fluoride ions was higher on the surface of MS-PVDF-Boehmite than that of MS and MS-Boehmite due to the large loading amount of boehmites which have a strong affinity toward fluoride ions. The Freundlich isotherm also gives an important factor of 1/n as an indicator of adsorption preference. When the *1/n* value ranges from 0 to 1, the adsorption process is favorable, and a smaller value suggests a more heterogeneous surface of the adsorbent and nonlinear isotherm [43]. On the other hand, if this value is greater than 1, the adsorption process becomes unfavorable. The *1/n* values of MS, MS-Boehmite, and MS-PVDF-Boehmite were calculated to be 0.590, 0.562, and 0.223, respectively, which are all less than 1, implying that the adsorption process of fluoride ions on their surface was favorable. Those results concurred well with the experimental results that MS-PVDF-Boehmite presented a much higher fluoride removal efficiency than MS and MS-Boehmite. Then, the error functions were compared to ensure the reliability of the isotherm modeling results. In all cases of MS, MS-Boehmite, and MS-PVDF-Boehmite, R^2^ values were close to 1, and χ^2^ and SAE values were also relatively low in the Freundlich isotherm model compared to the Langmuir model. This result indicates that the Freundlich isotherm is more appropriate for describing the adsorption process, and thus the multilayer adsorption is dominant for fluoride ions onto MS, MS-Boehmite, and MS-PVDF-Boehmite.

### 3.6. Adsorption Kinetics

The effect of adsorption time on removal efficiency of fluoride ions was investigated by conducting adsorption experiments at 20 mg L^−1^ of fluoride ions with varying adsorption times from 10 to 1440 min (Figure 7a). Within 30 min of adsorption, MS-PVDF-Boehmite exhibited a rapid adsorption process compared to MS and MS-Boehmite. A total of 62% of fluoride ions were removed by MS-PVDF-Boehmite, whereas only 2% and 25% of fluoride ions were removed by MS and MS-Boehmite (Figure 7a). At the equilibrium state (after 1440 min of adsorption time), MS, MS-Boehmite, and MS-PVDF-Boehmite exhibited removal efficiencies of 44%, 53%, and 79%, respectively. The presence of positively charged boehmites on the surface of MS-PVDF-Boehmite facilitated the adsorption of fluoride ions through electrostatic interactions, leading to formation of a strong bonding between them. As a result, a higher boehmite content on the surface of MS-PVDF-Boehmite leads to a faster adsorption process and higher adsorption capacity at equilibrium than MS and MS-Boehmite.

The pseudo-first-order and pseudo-second-order kinetic models were employed to investigate the adsorption process and determine the kinetic parameters based on the experimental adsorption data at different contact times. The pseudo-first-order kinetic model is typically applied to reversible reactions where an equilibrium is established between the liquid and solid phases, while the pseudo-second-order model assumes that the rate-determining step involves chemisorption with valence forces through electron sharing or exchange between the adsorbent and adsorbate [44,45]. Kinetic curves and parameters from the experimental adsorption data are shown in Figure 7b,c and Table 3, respectively. The linearized forms of the pseudo-first-order and pseudo-second-order kinetic equations are given by Equations (5) and (6), respectively:(5)$$ln\left(q_{e}-\;q_{t}\right)=ln\;q_{e}-\;k_{1}t\;\;$$
(6)$$\frac{t}{q_{t}}=\frac{1}{k_{2}q_{e}^{2}}+\frac{1}{q_{e}}\;t\;\;$$

Here, *q_t_* represents the adsorption capacity of fluoride at contact time (mg g^−1^), while *q_e_* represents the adsorption capacity at the equilibrium state. The rate constants for the pseudo-first-order and pseudo-second-order models are denoted as *k*_1_ (min^−1^) and *k*_2_ (g mg^−1^·min^−1^), respectively, while *t* (min) indicates the contact time.

The values of error functions in Table 3 indicate that the pseudo-second-order model provided a better fit for MS, MS-Boehmite, and MS-PDVF-Boehmite based on the high R^2^ values (0.999) and markedly lower values of SAE and χ^2^ compared to the pseudo-first-order model. Furthermore, the equilibrium adsorption capacity (*q_e,cal_*) from the pseudo-second-order model was determined to be 2.578, 2.322, and 1.007 mg g^−1^ for MS, MS-Boehmite, and MS-PVDF-Boehmite, respectively, which were closely matched with the experimentally obtained adsorption capacity (*q_e,exp_*). Those results also implied that the adsorption process of fluoride ions on MS, MS-Boehmite, and MS-PVDF-Boehmite was well described with the pseudo-second-order kinetic model rather than the pseudo-first-order kinetic model. The kinetic analysis further confirmed that fluoride ions were dominantly removed through the strong electrostatic interactions between negatively charged fluoride ions and positively charged boehmites.

To further investigate the rate-determining step during the adsorption process, the experimental adsorption data were plotted by the Weber–Morris intraparticle diffusion model, and the intradiffusion curves and parameters are shown in Figure 7d and Table 3, respectively. The equation of this diffusion model is expressed as follows in Equation (7):(7)$$q_{t}=k_{id}t^{1/2}+C$$

The intraparticle diffusion model incorporates parameters such as *k_id_*, *C*, and *q_t_*, which represent the intraparticle diffusion rate constant (mg g^−1^∙min^−1/2^), the thickness of the boundary layer (mg g^−1^), and the adsorption capacity at a given contact time (mg g^−1^). According to the Weber–Morris model, if the plot of the adsorption data follows a straight line, it suggests the intraparticle diffusion process is rate-controlling. Conversely, if the plot passes through the origin, it implies that the intraparticle diffusion is the rate-determining step [46]. The intradiffusion curves were roughly divided by two straight lines with different slopes and none of lines passed through the origin of graph (Figure 7d). This result implied that the intraparticle diffusion was not solely the rate-determining step, and there was influence of boundary layer diffusion. Considering the slopes of two straight lines, the main rate-determining step was intraparticle diffusion because its slope is smaller than that of boundary layer diffusion.

### 3.7. Adsorption Performance According to Flow Rate

For the practical application of the MS-PVDF-Boehmite composite membrane, its adsorption performance needs to be evaluated with flowing wastewater containing fluoride ions with different flow rates. Figure 8a showed the experimental setup of the adsorption test with flowing wastewater. MS-PVDF-Boehmite was cut to fit into a syringe (π × 0.6^2^ × 4) and used as a cartridge to remove fluoride ions from flowing wastewater. After putting 5 mL of a 20 mg L^−1^ solution of fluoride ions into a syringe, the flow rate was controlled with a syringe pump at 30, 70, and 110 mL h^−1^, and the concentration of fluoride ions in the treated water was measured. This filtration process was repeated for several cycles until the concentration of fluoride ions was lower than the WHO standard (1.5 mg L^−1^).

At a flow rate of 30 mL h^−1^, the filtered solution through MS-PVDF-Boehmite showed 0.9 mg L^−1^ of fluoride ions after four cycles of filtration, while those through MS and MS-Boehmite showed 6.2 and 4.9 mg L^−1^ (Figure 8b). However, the filtered solution was slightly opaque (an inset of Figure 8b), and it implied that boehmites were partially detached into the filtered water during the repeated filtration processes owing to a prolonged contact with wastewater with a low flow rate. When the flow rate increased to 70 mL h^−1^, the removal efficiency of fluoride ions was not changed regardless of the composite membranes, but the filtered solutions through them became clear. This result signified that the increase in flow rate prevented a detachment of boehmites from MS-PVDF-Boehmite during the filtration process without deterioration of its adsorption performance. However, with a further increase in flow rate to 110 mL h^−1^, the removal efficiency of MS, MS-Boehmite, and MS-PVDF-Boehmite for fluoride ions declined sharply to 10.8, 8.2, and 4.0 mg L^−1^, respectively, because the contact time of wastewater with the composite membranes decreased.

## 4. Conclusions

A composite membrane for the removal of fluoride ions was developed by a simple dip-coating process. **(R1-4)** The characterization results suggested that the thermal and mechanical properties of MS were enhanced with a loading of boehmites, and the loading amount of boehmite increased greatly with a PVDF adhesive layer. Then, the prepared composite membranes were applied to the removal of fluoride ions by two different processes such as incubation and filtration. This study found that the MS-PVDF-Boehmite showed the highest performance to remove fluoride ions through both processes. At low concentrations below 10 mg L^−1^, fluoride ions were completely removed with MS-PVDF-Boehmite within 1 h of incubation. At a high concentration of 20 mg L^−1^, its removal efficiency was 78.6% and it was maintained to 51.8% even at 80 mg L^−1^ after 24 h of incubation. The experimental results were applied to Langmuir and Freundlich adsorption isotherms as well as kinetic analysis to study the adsorption characteristics of the prepared composite membranes. **(R2-5)** The Freundlich adsorption isotherm and pseudo-second-order kinetic model were found to be the best-fitted models for MS-PVDF-Boehmite. Furthermore, the Weber–Morris intraparticle diffusion model indicated that the diffusion rate was not solely affected by intraparticle diffusion, and it was also influenced by boundary layer diffusion. The modeling studies revealed that the adsorption of fluoride ions on MS, MS-Boehmite, and MS-PVDF-Boehmite occurred through chemical interaction with valance forces between positively charged boehmite and negatively charged boehmite. Finally, the composite membranes were inserted into a syringe as a cartridge, and adsorption performance was evaluated at varying flow rates. After four cycles of filtration at a flow rate of 70 mL h^−1^, the concentration of fluoride ions fell to 0.9 mg L^−1^ with MS-PVDF-Boehmite, which is below the WHO standard (1.5 mg L^−1^). We believe that MS-PVDF-Boehmite can be a simple, efficient, and practical tool for the removal of fluoride ions from wastewater owing to its simple fabrication process, cost-effectiveness, and high performance.

## Figures and Tables

**Figure 1 polymers-15-02916-f001:**
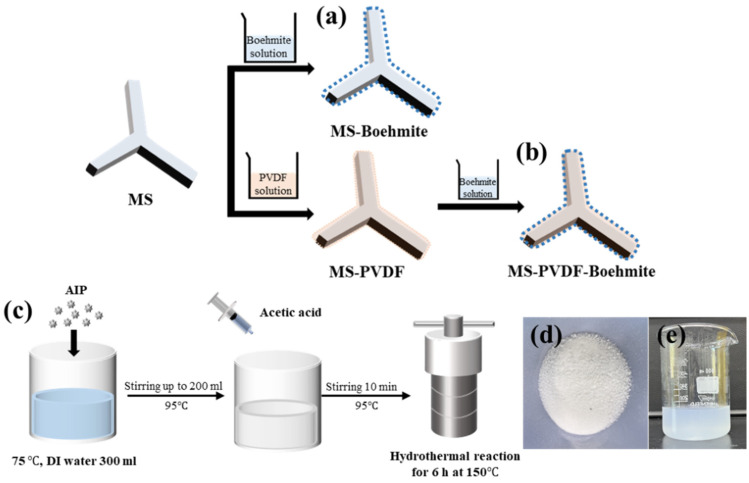
Schematic diagram of preparation of (**a**) MS-Boehmite and (**b**) MS-PVDF-Boehmite composite membranes. (**c**) A schematic illustration of boehmite synthesis. Photographs of (**d**) the dried and (**e**) suspended boehmites in water.

**Figure 2 polymers-15-02916-f002:**
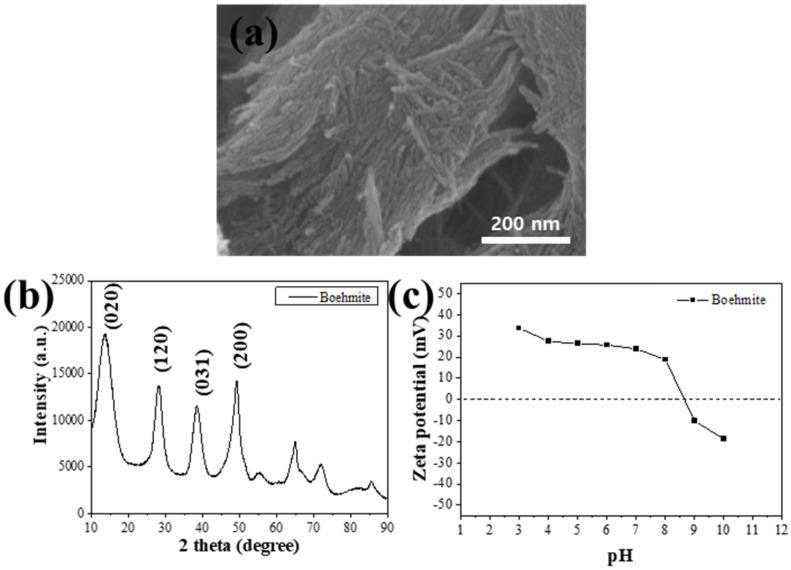
(**a**) SEM image and (**b**) XRD pattern of the synthesized boehmite. (**c**) Zeta potential values of boehmites suspended in water with varying pH conditions.

**Figure 3 polymers-15-02916-f003:**
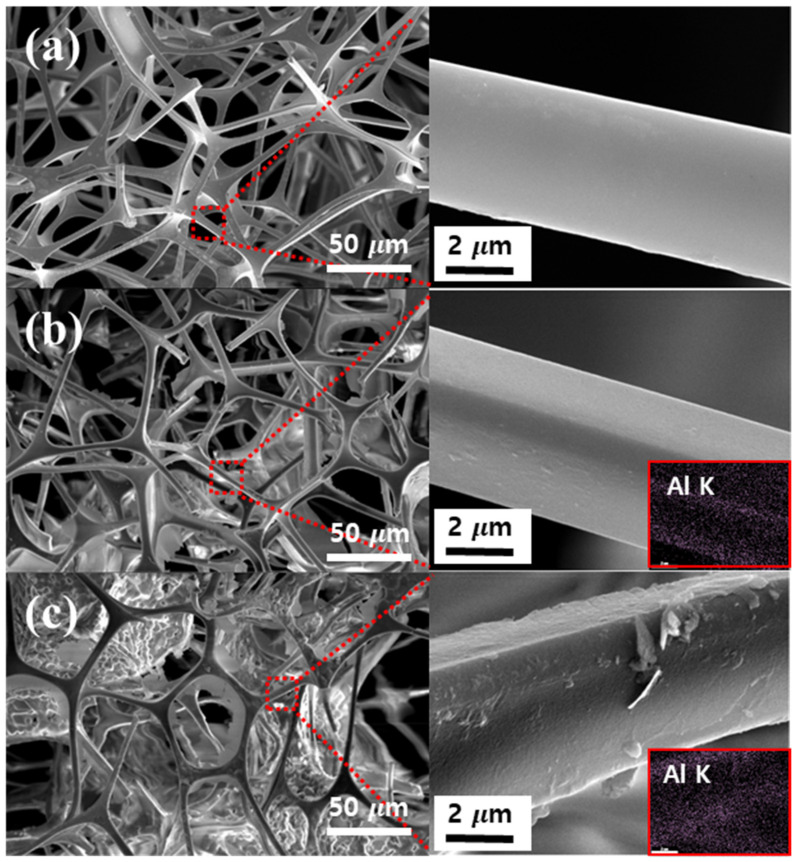
SEM and EDX images of (**a**) MS, (**b**) MS-Boehmite, and (**c**) MS-PVDF-Boehmite.

**Figure 4 polymers-15-02916-f004:**
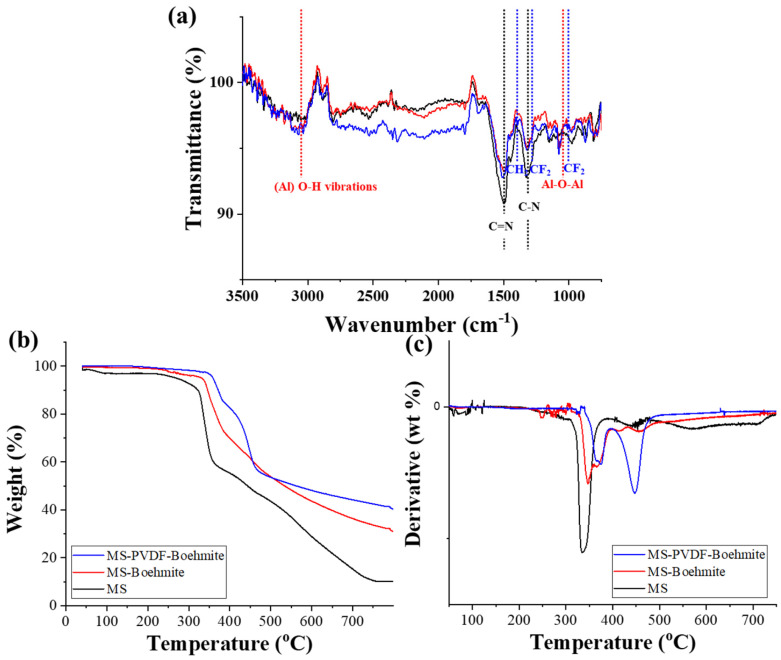
Characterization of MS, MS-Boehmite, and MS-PVDF-Boehmite: (**a**) FT-IR, (**b**) TGA, and (**c**) DTA.

**Figure 5 polymers-15-02916-f005:**
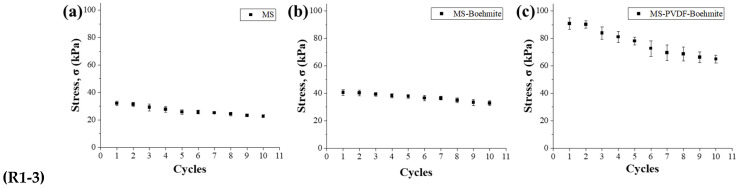
Stress according to the number of repetitions of compression of (**a**) MS, (**b**) MS-Boehmite, and (**c**) MS-PVDF-Boehmite with a constant strain rate of 10 mm min^−1^. The insets show the compression test images of the MS, MS-Boehmite, and MS-PVDF-Boehmite samples.

**Figure 6 polymers-15-02916-f006:**
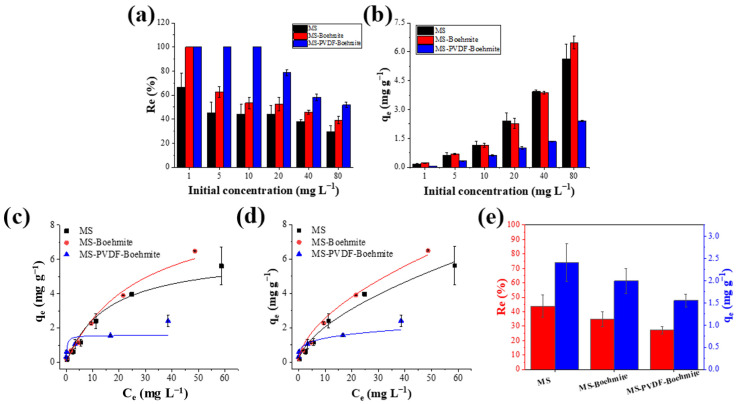
Effect of initial fluoride ion concentration on (**a**) adsorption efficiency and (**b**) adsorption capacity. (**c**) Langmuir and (**d**) Freundlich isotherm models to investigate the adsorption process of fluoride ions on MS, MS-Boehmite, and MS-PVDF-Boehmite. **(R2-4)** (**e**) The removal efficiency and adsorption capacity of MS, MS-Boehmite, and MS-PVDF-Boehmite for fluoride ions by incubation process.

**Figure 7 polymers-15-02916-f007:**
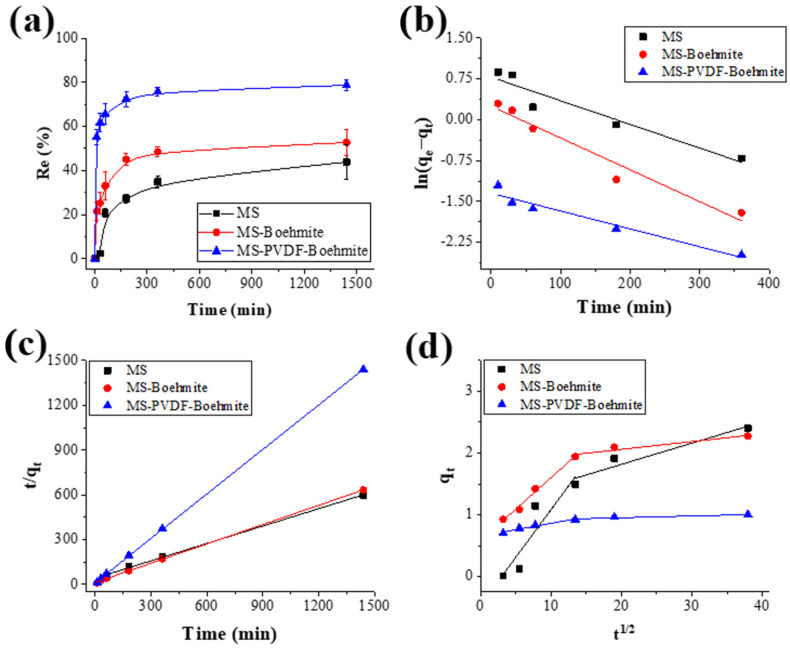
(**a**) Effect of contact time for fluoride adsorption onto adsorbents, (**b**) pseudo-first-order model, (**c**) pseudo-second-order model, and (**d**) Weber–Morris intraparticle diffusion model for adsorption kinetic study of fluoride ions.

**Figure 8 polymers-15-02916-f008:**
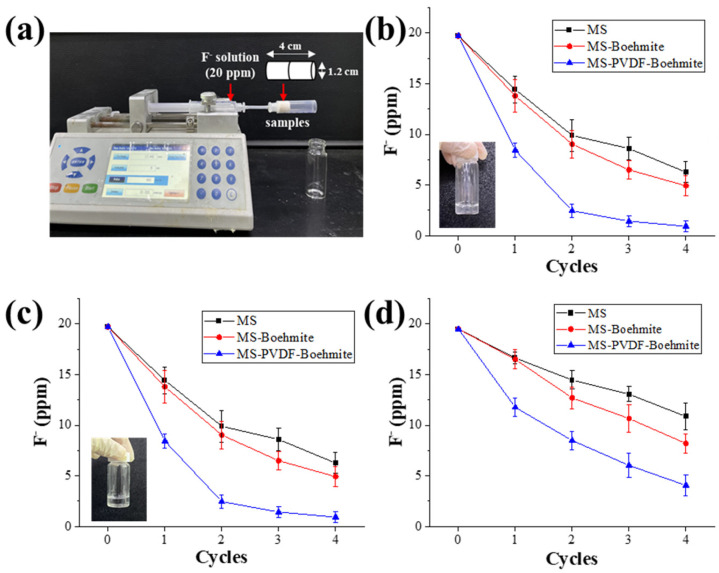
Comparison of mg L^−1^ concentration before and after filtering test of composite membrane for removing fluoride in different flows: (**a**) experimental setup, (**b**) 30 mL h^−1^, (**c**) 70 mL h^−1^, and (**d**) 110 mL h^−1^.

**Table 1 polymers-15-02916-t001:** Error functions for estimation of nonlinear regression models.

Error Function	Equation
Coefficient of determination (R^2^)	$\frac{\sum_{i=1}^{n}{\left[\left(q_{e,meas}-\bar{q_{e,cal}}\right)\right]}_{i}^{2}}{\sum_{i=1}^{n}{\left[{\left(q_{e,meas}-\bar{q_{e,cal}}\right)}^{2}+{\left(q_{e,meas}-q_{e,cal}\right)}^{2}\right]}_{i}}$
Nonlinear chi-square (χ^2^)	$\overset{n}{\underset{i=1}{\sum }}{\left[\frac{{\left(q_{e,meas}-q_{e,cal}\right)}^{2}}{q_{e,meas}}\right]}_{i}$
Sum of absolute errors (SAE)	$\overset{n}{\underset{i=1}{\sum }}{\left|q_{e,meas}-q_{e,cal}\right|}_{i}$

**Table 2 polymers-15-02916-t002:** Parameters calculated from the Langmuir and Freundlich isotherm models.

Case	Isotherm Model	The Calculated Parameters			Error Functions		
	Langmuir	*q_max_*	*K_L_*		R^2^	χ^2^	SAE
MS		6.36	0.064		0.943	0.760	2.688
MS-Boehmite		9.47	0.036		0.920	0.253	1.345
MS-PVDF-Boehmite		1.58	3.655		0.927	0.674	1.605
	Freundlich	*K_F_*	*1/n*		R^2^	χ^2^	SAE
MS		0.528	0.590		0.978	0.454	2.089
MS-Boehmite		0.704	0.562		0.992	0.150	1.325
MS-PVDF-Boehmite		0.837	0.223		0.993	0.140	0.733

**Table 3 polymers-15-02916-t003:** Parameters calculated from the pseudo-first-order, pseudo-second-order, and intraparticle diffusion kinetic models.

Case	*q_e,exp_*	Pseudo-First-Order Model							
		*k* _1_	*q_e,cal_*	R^2^	SAE	χ^2^			
MS	2.406	0.004	2.188	0.917	1.353	1.381			
MS-Boehmite	2.276	0.006	1.283	0.956	3.695	2.322			
MS-PVDF-Boehmite	1.001	0.003	0.260	0.946	2.484	1.682			
	*q_e,exp_*	**Pseudo-second-order** **Model**							
		*k* _2_	*q_e,cal_*	R^2^	SAE	χ^2^			
MS	2.406	0.002	2.578	0.999	0.420	0.057			
MS-Boehmite	2.276	0.015	2.322	0.999	0.582	0.158			
MS-PVDF-Boehmite	1.001	0.057	1.007	0.999	0.334	0.079			
		**Intraparticle diffusion** **Model**							
		*k_id_* _1_	*k_id_* _2_	R_1_^2^	SAE_1_	χ_1_^2^	R_2_^2^	SAE_2_	χ_2_^2^
MS		0.154	0.034	0.848	0.810	0.741	0.935	0.258	0.002
MS-Boehmite		0.102	0.013	0.990	0.136	0.005	0.936	0.093	0.002
MS-PVDF-Boehmite		0.021	0.003	0.960	0.063	0.001	0.891	0.028	0.001

## Data Availability

Not applicable.

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
