# Peer review of "Composite Membrane Based on Melamine Sponge and Boehmite Manufactured by Simple and Economical Dip-Coating Method for Fluoride Ion Removal"

_polymers, 2023, doi:10.3390/polym15132916_

Round 1
Reviewer 1 Report
The authors of the manuscript described " Composite membrane based on melamine sponge and Boehmite manufactured by simple and economical dip coating method for fluoride ion removal" in detail. The experimental samples are fully characterized, and the innovation point is outstanding. I think the author has done a good job and the paper will attract broad readers if published. However, a few shortcomings are still there, and I suggest minor review at this stage. I have the following concerns about this manuscript.
1. In Section 2.1, describe the purity of the materials used.
2. In Fig.2a, Boehmite is negatively charged when PH=8-10. However, the author only analyzed the reason of PH=3-8 in the manuscript, please add the reason why Boehmite has negative electricity in the manuscript.
3. Why is there no error bar in Fig.5? Please add error bar to maintain the accuracy of data.
4. The conclusion is a bit long, the author should not describe the experimental process in large paragraphs when writing the conclusions, instead, giving a summary of experiment conclusions is encouraged.
Please check the formatting of the references and grammar issues in the manuscript.
Author Response
Responses to Reviewer #1
We thank the Reviewer for the comments and suggestions. Summarized below are our responses to the Reviewer’s specific concerns.
Comment 1: In Section 2.1, describe the purity of the materials used.
Response 1: The purity information of materials was added in the revised manuscript.
Comment 2: In Fig.2f, Boehmite is negatively charged when pH=8-10. However, the author only analyzed the reason of pH=3-8 in the manuscript, please add the reason why Boehmite has negative electricity in the manuscript.
Response 2: It was explained in the revised manuscript as follow.
The negative charge of boehmite was caused by the increase in the number of OH- groups with the increase in pH, and the decrease in zeta potential value [37].
Comment 3: Why is there no error bar in Fig.5? Please add error bar to maintain the accuracy of data.
Response 3: It was corrected in the revised manuscript.
Comment 4: The conclusion is a bit long, the author should not describe the experimental process in large paragraphs when writing the conclusions, instead, giving a summary of experiment conclusions is encouraged.
Response 4: The conclusion was rewritten according to reviewer’s comment in the revised manuscript.
Reviewer 2 Report
Reviewer comments
Polymers-2422930
Composite membrane based on melamine sponge and boehmite manufactured by simple and economical dip coating method for fluoride ion removal
1. Abstract: Needs improvement; I suggest you may rewrite the abstract to include some info on composites, membrane filtration, melamine sponge, why your work is important, mention the characterization techniques, results and the models .
2. Introduction: Needs improvement: I suggest adding literature on composite membranes and applications, use of such membrane in removing fluoride ions ; what is not explained in previous research work, why your research is important and then add what and why you want to do through your current research. Use one unit consistently either ppm or mg/L in the manuscript.
3. Methods: Boehmite synthesis and Figure 2a, 2b, 2c may be moved under methods
4. Results and Discussion: Add references for the crystal planes and 2-theta values of Boehmite. Add Figure S1 in the manuscript and remove supplementary information from the submission. The experimental data and the discussions are good and adequate.
5. Conclusion: Add discussions on the models and what best explains the results
6. Overall: The manuscript has good data and discussion; but it needs to be more focused .
Author Response
Responses to Reviewer #2
We thank the Reviewer for the comments and suggestions. Summarized below are our responses to the Reviewer’s specific concerns.
Comment 1: Abstract: Needs improvement; I suggest you may rewrite the abstract to include some info on composites, membrane filtration, melamine sponge, why your work is important, mention the characterization techniques, results and the models.
Response 1: In the revised manuscript, the characterization and importance of present study was pointed as follow.
From the physical, chemical, thermal, morphological, and mechanical analysis of present materials, this study provides MS-PVDF-Boehmite composite filter materials that is suitable for fluoride removal applications due to its simple fabrication process, cost-effectiveness, and high performance.
Comment 2: Introduction: Needs improvement: I suggest adding literature on composite membranes and applications, use of such membrane in removing fluoride ions; what is not explained in previous research work, why your research is important and then add what and why you want to do through your current research. Use one unit consistently either ppm or mg/L in the manuscript.
Response 2: Introduction was revised by adding literatures on composite membranes and importance of present study. The unit of concentration was unified as a mg L-1.
If it is difficult to use this adsorbent independently, it may contribute to adsorption by immobilizing it on a membrane [19,20]. Materials for the membrane include polyacrylonitrile (PAN), polyvinylidene fluoride (PVDF), and polytetrafluoroethylene (PTFE) [21-23]. As for related studies, a separator that adsorbs copper ions by grafting PAMAM on the surface of a PVDF membrane has been reported [24], and a MOF membrane for removing Cd and Zn prepared by electrospinning of Zr-based MOF-808 and hydrophilic PAN has been reported [25]. These materials are suitable for aqueous applications and exhibit physical and chemical stability. However, we studied boehmite and sponge-based composite membranes, including higher adsorption capacity and superior mechanical properties, in addition to the advantages of previously reported composite membranes.
In this work, a composite membrane for removal of fluoride ions was developed by a simple dip coating process. The results indicated that MS-PVDF-Boehmite composite membrane is also an effective tool to purify a wastewater contaminated with fluoride ions.
Comment 3: Methods: Boehmite synthesis and Figure 2a, 2b, 2c may be moved under methods.
Response 3: Figure 2a-c was moved to Figure 1b-d in the revised manuscript.
Comment 4: Results and Discussion: Add references for the crystal planes and 2-theta values of Boehmite. Add Figure S1 in the manuscript and remove supplementary information from the submission. The experimental data and the discussions are good and adequate.
Response 4: Figure S1 was moved to Figure 6e in the revised manuscript.
Comment 5: Conclusion: Add discussions on the models and what best explains the results.
Response 5: In the revised manuscript, discussions and best-fitted models for adsorption isotherm and kinetic were presented as follows.
The Freundlich adsorption isotherm and pseudo-second order kinetic model were found to be the best-fitted models for MS-PVDF-Boehmite. Furthermore, Weber-Morris intraparticle diffusion model indicated that the diffusion rate was not solely affected by intraparticle diffusion, and it was also influenced by boundary layer diffusion.
Comment 6: Overall: The manuscript has good data and discussion; but it needs to be more focused.
Response 6: We appreciated reviewer’s comment, and the manuscript was revised for better readership.
Round 2
Reviewer 2 Report
I appreciate that the authors considered my comments and revised the manuscript accordingly. The revised manuscript is good. Please note reference 17 has typos.